# Effects of Room Temperature Stretching and Annealing on the Crystallization Behavior and Performance of Polyvinylidene Fluoride Hollow Fiber Membranes

**DOI:** 10.3390/membranes10030038

**Published:** 2020-02-29

**Authors:** Yuanhui Tang, Yakai Lin, Hanhan Lin, Chunyu Li, Bo Zhou, Xiaolin Wang

**Affiliations:** 1College of Chemistry and Environmental Engineering, China University of Mining and Technology, Beijing 100083, China; tangyuanhui@126.com (Y.T.); 18810922797@163.com (C.L.); 2Beijing Key Laboratory of Membrane Materials and Engineering, Department of Chemical Engineering, Tsinghua University, Beijing 100084, China; xl-wang@tsinghua.edu.cn; 3Beijing Scinor Membrane Technology Co. Ltd., Bejing, 100083, China; 4Sinopec Economics & Development Research Institute, Beijing 100029, China; linhanhan050505@163.com; 5Small and medium-sized enterprise service center of Quzhou, Zhejiang 324000, China; zhoubo09@126.com

**Keywords:** poly(vinylidene fluoride), hollow fiber microporous membrane, room temperature stretching, annealing, crystallization

## Abstract

A treatment consisting of room temperature stretching and subsequent annealing was utilized to regulate the morphology and performance of polyvinylidene fluoride (PVDF) hollow fiber membranes. The effects of stretching ratios and stretching rates on the crystallization behavior, morphology, and performance of the PVDF membranes were investigated. The results showed that the treatment resulted in generation of the β crystalline phase PVDF and increased the crystallinity of the membrane materials. The treatment also brought about the orientation of the membrane pores along the stretching direction and led to an increase in the mean pore size of the membranes. In addition, as the stretching ratio increased, the tensile strength and permeation flux were improved while the elongation at break was depressed. However, compared to the stretching ratio, the stretching rate had less influence on the membrane structure and performance. In general, as the stretching ratio was 50% and the stretching rate was 20 mm/min, the tensile strength was increased by 36% to 7.47 MPa, and the pure water flux was as high as 776.28 L/(m^2^·h·0.1bar), while the mean pore size was not changed significantly. This research proved that the room temperature stretching and subsequent annealing was a simple but effective method for regulating the structure and the performance of the PVDF porous membranes.

## 1. Introduction

Polyvinylidene fluoride (PVDF), as a semi-crystalline polymer with good mechanical properties, excellent chemical resistance, and thermal stability, has been widely applied in polymeric membrane fabrication. PVDF microporous membranes have been widely employed in the treatment of municipal sewage and industrial wastewater [1,2,3]. Depending on its different chain conformations of trans (T) and gauche (G), semi-crystalline PVDF possesses many complicated structures with at least four crystalline polymorphs, including α (TGTG’), β (TTTT), γ (T_3_GT_3_G’), and δ (polarized α) phases [4]. The permeability and mechanical performance of the membranes are closely related to the crystalline phase-type and crystallization behavior of PVDF [4,5]. In recent years, many efforts have been made to study the influence of PVDF crystallization on membrane structure and performance. The results have shown that the mechanical performance of the PVDF membranes could be improved by increasing the membrane crystallinity as well as the β crystalline phase content [6,7,8]. Ma et al. used nanoparticles as additives and prepared PVDF membranes. The results demonstrated that the addition of the nanoparticle montmorillonite improved the membrane structure due to a transition of the membrane crystalline phase partly from α to β [9]. 

Stretching and annealing are simple and effective methods for the preparation and modification of PVDF materials. On the basis of the investigation on the preparation and modification of the PVDF fibers and films, it has been found that the PVDF crystallization behavior can be regulated to realize a crystalline phase transition from α to β and increase the crystallinity [10,11,12]. Zhu et al. found that by a cooperative stretching process of PVDF films deposited on stretchable poly (vinyl alcohol) substrates, the PVDF molecules could be oriented along the direction of stretching and the nonpolar α phase was converted to electroactive β and γ phases with an increase of stretching ratio from 1 to 3. In addition, both the stretching rate and stretching temperature had effects on this process [10]. Lei et al. studied the influences of the stretching rate and stretching temperature on the crystalline phase transition and microporous structure of the PVDF thin films. The results indicated that there was lamella crystal separation, crystalline phase transition, and recrystallization simultaneously happening during the stretching process [11,12]. Beaume et al. found that annealing at 140 °C resulted in an increase of the β crystalline phase fraction and facilitated further crystallization of PVDF materials [13].

Thermally induced phase separation (TIPS) method is a simple process with strong controllability and few parameters for PVDF microporous membrane preparation. In the membrane formation process, based on the compatibilities between PVDF and the solvents, the PVDF-solvent system can undergo a thermally induced liquid–liquid phase separation (TIPS (L–L)) to form membranes with bicontinuous cross-section structure or a thermally induced solid–liquid phase separation (TIPS (S–L)) to form membranes with spherulitic cross-section structure [14]. In recent years, a great deal of effort has been given to prepare PVDF membranes with a bicontinuous microporous structure and high performance via the TIPS (L–L) process [9,15,16,17]. The results have shown that generally α crystalline phase was mainly formed during the TIPS (L–L) process rather than the β crystalline phase, which restricted the mechanical strength of the resultant PVDF membranes. Therefore, some people have tried to use a post-treatment consisting of stretching and annealing to improve the PVDF membranes. Lloyd et al. studied the effects of different stretching conditions on pore structure and membrane surface roughness since they thought that higher surface roughness could promote the antifouling property of the membranes [18,19]. Whereas, they did not study the influence of the stretching on the membrane permeability and mechanical properties. Lee et al. conducted full and comprehensive research to investigate the effects of the stretching rate, stretching ratio, stretching temperature, and annealing temperature on the permeability and mechanical properties of the PVDF microporous membranes [20]. The results showed that the stretching was able to improve the porosity and crystallinity of the membranes, while the annealing temperature of around 130 °C was more appropriate for the membrane performance improvement. However, this study focused on the PVDF membranes with mainly spherulitic structure rather than a bicontinuous microporous structure obtained via the TIPS (L–L) process. Lee et al. also prepared microporous PVDF membranes via the TIPS and stretching methods [21]. They found that the stretching ratio positively affected the permeability and porosity but decreased the fiber strain and stretching the fibers up to 40% was not enough to induce any detectable β phase crystals.

Therefore, to explore more methods for improving the permeability and mechanical strength of the PVDF membranes with bicontinuous cross-section structure prepared via the TIPS (L–L) process, herein the PVDF hollow fiber membranes were modified by a treatment consisting of room temperature stretching and subsequent annealing at a high temperature to release stress. The effects of the stretching ratio and the stretching rate on the crystallization behavior, membrane structure and performance were investigated to explore the optimum operation condition and the mechanism for the membrane improvement. It should be noted that, herein, the stretching ratio indicates the length ratio of the stretched membranes to the original ones.

## 2. Materials and Methods 

In this work, the PVDF hollow fiber microporous membranes were prepared via the TIPS (L–L) process according to the method described in the literature [15,22] as the PVDF concentration was 26 wt.%. The inner and outer diameters of the membranes were approximately 0.75 mm and 1.3 mm, respectively. The ethanol and the water used were analytical grades.

### 2.1. Preparation of the Membrane Samples

The PVDF hollow fiber membranes were fixed on a universal testing machine (Shimadzu AGS-J 20N, Kyoto, Japan) and, then, stretched to a certain length at the room temperature. Subsequently, the samples were kept at 145 °C which was near the onset crystallization temperature of PVDF (around 150 °C) under the strain for 15 min annealing to release the stress. The stretching rates used in this work were 20, 40, and 80 mm/min, while the stretching ratios were 20, 50, 80, and 100%, because the membrane would be broken as the stretching ratio was higher than 100%. The preparation conditions of the membrane samples are shown in Table 1.

### 2.2. Characterization of Membrane Samples

The chemical composition of the membrane samples was characterized using an attenuated total reflectance (ATR)-Fourier transform infrared (FTIR) spectroscopy (Nicolet 6700, Thermo Fisher Scientific Inc., Waltham, MA, USA). Zinc selenide (ZnSe) was applied as an internal reflection element, and each spectrum was captured by 64 averaged scans at a resolution of 4 cm^−1^.

Wide angle X-ray diffraction (WAXD) was conducted in a Bruker D8-Advance diffractometer (Cu K_α_ radiation, 40 kV and 40 mA) to identify the crystalline phase of the membranes. The scanning angle ranged from 16° to 28° with a scanning velocity of 10 °/min. 

The crystallization temperature and crystallinity were determined by a differential scanning calorimetry (DSC, TA Q200). The membrane samples were heated to 220 °C at 10 °C/min, and then cooled to 40 °C. The melting data were used to draw the DSC curves and calculate the PVDF crystallinity *Xc* by the following equation,
(1)$$X_{c}=\frac{△H_{f}}{△H_{f}^{*}}\times 100\%$$
where $△H_{f}^{*}=104.5\;J/g$ [23] was the melting enthalpy for a 100% crystalline sample of PVDF, and $△H_{f}$ was the melting enthalpy of the PVDF membrane measured by the DSC. The morphology of the PVDF hollow fiber membrane samples was examined using a scanning electron microscope (SEM, JEOL JSM7401). The membrane samples were fractured in liquid nitrogen and coated with platinum. The SEM with the accelerating voltage of 3.0 kV was used to examine the cross-section and surface morphology of the membranes.

The mechanical strength of the hollow fiber membranes was measured via a universal testing machine (Shimadzu AGS-J 20N) equipped with a 5 kg load cell. Each sample was stretched at 40 mm/min. The initial distance between the clamps was 100 mm. 

The static contact angles of the membrane samples were measured by a contact angle meter (OCA20, Dataphysics Co., Ltd., Filderstadt, Germany) at room temperature. 

The bubble point test was applied to measure the mean pore size. The membrane sample was soaked in an ethanol bath. Then nitrogen gas was pressed inside-out from the hollow fiber membrane sample under certain pressure and the pressure was increased gradually. When there were continuous bubbles emerging on the membrane surface, the pressure (Pa) was recorded as *P*. The mean pore size (MPS) was calculated according to the following equation,
(2)$$\;MPS=\frac{2\sigma }{P}$$
where *σ* was the surface tension of the ethanol at the tested temperature. 

The pure water flux of the hollow fiber membrane samples was determined using a self-made dead-end filtration under the transmembrane pressure of 0.1 bar. Equation (3) is usually applied to calculate the pure water flux (PWF) data.
(3)$$\mathrm{PWF}=\frac{Q}{\pi \mathrm{DLt}}$$
where Q, D, L, and t represented the pure water permeation amount (m^3^), the outer diameter of the hollow fiber membrane (m), the effective length of the hollow fiber membrane (m), and the permeation time (h), respectively. However, since in this work the hollow fibers have thick walls, the calculated PWF data obtained by the outer surface area would be quite different from that using the inner surface area. Therefore, according to the literature [24], the PWF (unit: L·m^−2^·h^−1^·(0.1bar)^−1^) was calculated according to the following equation,
(4)$$\mathrm{PWF}=\frac{Q}{2\pi \mathrm{Lt}\left(r_{2}-r_{1}\right)}\ln\left(\frac{r_{2}}{r_{1}}\right)$$
where $r_{2}$ and $r_{1}$ represented the outer radius and inner radius of the hollow fiber membrane (m), the effective length of the hollow fiber membrane (m), and the permeation time (h), respectively. It should be noted that the PWF data were converted to that tested at 25 °C to avoid the influence of the testing temperature.

All the above tests were repeated more than three times to eliminate the randomness.

## 3. Results and Discussion

When the PVDF hollow fiber membranes with the bicontinuous microporous structure are stretched, the polymer molecules are oriented along the stretching direction and, then, the structure of the molecular chains in the crystal lattice are changed, which facilitates the transformation of the crystalline phase and the pore structure. Annealing releases the residual stress and also promotes the recrystallization of the polymer chains. Here in this work, the effects of the stretching ratio and stretching rate on the crystallization behavior and performance of the PVDF hollow fiber microporous membranes are investigated.

### 3.1. Effect of the Stretching Ratio

Figure 1 shows the ATR-FTIR spectra (A) and the WAXD diffractograms (B) of the membrane samples stretched with different stretching ratios. As shown in Figure 1A, generally all the membrane samples show the characteristic α phase bands (796, 855, and 976 cm^−1^), which means the crystalline phase of all the membrane samples is still dominated by α, which agrees with the results of the previous literature [20]. As the stretch ratio increased from 0% to 100%, the intensity of the characteristic α phase bands does not change significantly, while that of characteristic β phase bands at 840 and 1275 cm^−1^ increases gradually, which suggests that the stretching and annealing treatment can result in an enhancement in β crystalline phase formation, and the β crystalline phase intensity increases with a rise in the stretching ratio. This trend can also be obtained from Figure 1B. The X-ray diffractograms of all the membrane surfaces are seen to have well-defined peaks at 2*θ* = 17.8°, 18.3°, 19.9°, and 26.6°, which correspond to (100), (020), (110), and (021) crystal plane of α-phase [25]. This phenomenon also demonstrates that α-phase holds a dominant position in the membranes before or after the stretching, although after the stretching and annealing, a shoulder peak at 2*θ* = 20.6° that corresponds to (110/200) plane appears and gradually becomes obvious as the stretch ratio is increased from 50% to 100%. Whereas the peak at 2*θ* = 20.6° is hardly observed in the Sample O and S1, which further confirms the influence of the external treatment on the crystal transformation. It should be noted that the intensity of the characteristic β phase is not obvious as the stretching ratio is lower than 50%, and becomes evident as the stretching ratio is getting higher, which agrees with the results of the literature [21] that “stretching the fibers up to 40% was not enough to induce any detectable β phase crystals”.

Table 1 and Figure 2 show the data of melting peak points, crystallinities, and the DSC curves of the membrane samples with different stretching ratios. It should be noted that the melting peak points of Table 1 are the peak points marked by the dashed box in Figure 2. The results in Table 1 illustrate that the melting peak points of the samples are hardly influenced by varying the stretching ratio, while the total crystallinity is significantly improved, which is consistent with the research in the literature [6]. As shown in the Figure 2 Sample O presents a single melting peak, while for Sample S1 to S4, a small melting peak near 157 °C appears and this peak seems to be independent of the stretching ratio. According to [26], this phenomenon should be attributed to a formation of a metastable phase of polymer chain crystallization of the amorphous region during annealing at 145 °C. At the same time, as the stretching ratio is increased, another melting peak appears on the right side of the dashed box and gradually shifts to a higher temperature. There are considerable controversies in the literature concerning the origin of the multiple melting peaks which takes place in many semi-crystalline polymers. Herein, according to the literature [9,27,28], this could be evidence of different crystalline forms of PVDF crystals because the melting peak temperature of the β phase PVDF crystal is higher than that of the α phase crystal [5]. 

The above studies have clarified that the room temperature stretching and annealing can cause the PVDF membrane samples to undergo crystalline phase transformation in the solid state, meanwhile, the extent of transformation and recrystallization is also able to be regulated by adjusting the stretching ratio. Next, the effect of the stretching ratio on the structure and performance of the hollow fiber membrane samples with bicontinuous structure is discussed. Figure 3 shows the SEM photographs of the inner surface, the outer surface, the cross-section (along the stretching direction), and the whole cross-section structures of the membrane samples obtained with different stretching ratios. We found that the treatment has a significant effect on the structure of the membrane samples. Generally, the cross-section structure (along the stretching direction) of the five samples all present bicontinuous, which implies that the stretching treatment cannot completely change the structure. As the stretching ratio is increased, the pores in the surface and the cross-section gradually become larger and oriented along the stretching direction. As the stretching ratio is 100%, the membrane pores (Figure 3S4(a–c)) have been significantly deformed and some small pores have merged into big pores due to the intense tensile force. Moreover, Table 2 lists the mean pore size and PWF of the membrane samples. According to Table 2, the mean pore size of the membrane sample is increased from 235 to 373 nm and the PWF is gradually improved from 457.01 to 1000.22 L/(m^2^·h·0.1bar). The results have shown that the external treatment can effectively increase the mean pore size and the PWF of the PVDF membrane samples, which agrees with the structure results.

Figure 4 shows the contact angles of the membrane samples with different stretching ratios. As shown in the figure, the contact angles of the membrane samples are gradually decreased by increasing the stretching ratio, indicating hydrophilicity of the membrane samples is enhanced correspondingly, which supports the PWF results of Table 2. According to the literature [18], the surface roughness and porosity also influence the contact angle values. Nevertheless, as shown in Figure 3, the outer surface structure of Sample S1 is similar to that of Sample O due to the fact that the stretching ratio is only 20%, whereas the contact angle of Sample S1 is still smaller than that of Sample O. Thus, the decrease in the contact angles should be attributed to the formation of the β crystalline phase of PVDF because normally the β crystalline phase has higher polarity [7,9]. The phenomenon suggests that increasing the stretching ratio results in an enhancement in the content of the β crystalline phase, which is in accordance with the results of Figure 1.

Figure 5 and Table 2 show the influence of stretching ratio on mechanical properties, including the stretching curve and the tensile strength of the membrane samples stretched with different stretching ratios. As the stretching ratio is increased, the tensile strength of the membrane samples is increased from 5.48 MPa of Sample O to 9.01 MPa of Sample S4. Figure 5 demonstrates that due to the stretching and annealing treatment, elastic modulus (slope of the stress-strain curves) of the membrane sample is almost unchanged while the elongation at break is depressed as the stretching ratio increased from 0% to 50%, indicating that the membrane samples gradually become stiff. This can be explained by the promotion in the crystallinity and these results are in accordance with the literatures [6,7,8], which indicates that increasing the membrane crystallinity and the β crystalline phase content can prompt the tensile strength of the PVDF membranes but decrease the membrane strain (the elongation at break). As the stretching ratio is increased to 80% and even 100%, the tensile strength of the membrane samples has risen to 7.83 and 9.01 MPa, while the elongation at break drops sharply, indicating that the membrane samples have become stiff and brittle. This is due to fact that at the large stretching ratio, the intense tensile force results in separation of the lamellas and the extension of the molecular chain, and thus defects such as macropores are formed in the membrane structure. As a result, the big change in the tensile strength and elongation at break is mainly related to the membrane structure damage when the stretching ratio is 80% and 100%. 

In general, the above results indicate that the stretching and annealing treatment can induce crystal phase transformation, recrystallization of the membranes, and a change in the membrane morphology and performance. According to the results of this section, it is easily found that the membrane structure has not changed significantly as the stretching ratio is 50%. Meanwhile, the mean pore size has increased by 23% and the tensile strength has increased by 20% to 6.56 MPa, while the PWF of the membrane is 776.28 L·m^−2^·h^−1^·(0.1bar)^−1^. It should be noted that the PWF of the PVDF membrane is normally below 1000 L·m^−2^·h^−1^·(bar)^−1^ which is much lower than the above result as the tensile strength is around 6.56 MPa, according to a current upper bound of the TIPS-prepared PVDF membranes from the literature [21]. Therefore, the stretching ratio of 50% is regarded as an effective and convenient method to significantly improve the PVDF membrane performance.

### 3.2. Effect of the Stretching Rate

Compared to the stretching ratio, the tensile force is also related to the stretching rate. This section further investigates the effect of the stretching rate on the crystallization behavior and performance of the membrane samples. Figure 6 shows the ATR-FTIR spectra of the membrane samples stretched with different stretching rates. Compared to Sample O, the ATR-FTIR spectrum of the membrane samples obtained by the stretching and annealing treatment all present characteristic β-phase bands at 840 and 1275 cm^−1^, indicating that the external treatment contributes to the crystal transformation of PVDF partially from α phase to β phase. This is consistent with the results of Figure 1. Theoretically, increasing the stretching rate always suppresses relaxation of the tensile force in the molecular chains during the stretching, thereby reducing the degree of crystal transformation. However, as shown in Figure 6, the intensity of the characteristic β phase bands at 840 and 1275 cm^−1^ does not vary visibly with the change in the stretching rate, which implies that the stretching rate has little effect on the formation of β crystal phase.

Figure 7 shows the DSC curves of Samples S5, S2, and S6 and Table 1 lists the melting peak points and total crystallinities of the four samples. As shown in Figure 7, the DSC curves of Samples S5, S2 and S6 look similar and they all have a small melting peak near 157 °C, as well as double melting peaks marked by the dashed box as compared with that of Sample O. Combining Figure 7 with Table 1, it is observed that with a rise in the stretching rate from 20 mm/min to 80 mm/min, there is not much difference in the total crystallinities, the melting points, and the DSC curves, indicating that changing the stretching rate from 20 mm/min to 80 mm/min does not obviously change the recrystallization and the total crystallinities, which agrees with Figure 6.

Figure 8 shows the structure of the outer surface, the inner surface, and the cross-section (along the stretching direction) of the membrane samples obtained by the different stretching rates. In general, it can be observed from the figure that varying the stretching rates does not have a significant influence on the membrane structure. For the lower stretching rate (20 mm/min), the effect of the tensile force lasts a little longer, which allows a longer relaxation time for the PVDF chain rearrangement and results in a higher porosity in the surface and cross-section of the samples. As the stretching rate is gradually increased to 40 mm/min, the porosity of the membrane structure slightly declines, and some big pores appear in the membrane structure since there is a shorter time for the polymer chain rearrangement. Moreover, as the stretching rate is increased to 80 mm/min, there are even some bigger holes that appear in the structure. In addition, Table 2 lists the mean pore size and PWF of the four samples, which presents that the PWF of Sample O is 457.01 L/(m^2^·h·0.1bar). As the stretching rate increased from 20 mm/min to 80 mm/min, the PWF of the membrane sample is decreased from 796.13 to 771.54 L/(m^2^·h·0.1bar), although the mean pore size is increased from 268.7 to 283.4 nm. This can be explained by the fact that the membrane structure obtained by the stretching rate of 20 mm/min is relatively more uniform, which brings about a higher permeability. For the membrane obtained by a higher stretching rate, the large holes contribute to the higher mean pore size, although they have no help from the PWF data. 

The mechanical performance of the membrane samples obtained with different stretching rates is shown in Figure 9 and Table 2. Compared to Sample O, the elongation at break is reduced a lot, which is consistent with the results of Section 3.1. When the stretching rate is lower (20 mm/min), the tensile strength of the membrane is increased to 7.47 MPa due to the longer relaxation time for the chain rearrangement. As the stretching rate is further increased to 40 mm/min and 80 mm/min, the stress-strain curves of the membrane samples almost coincide, while the tensile strength is decreased to 6.56 and 6.32 MPa. The decrease in the tensile strength should be attributed to the membrane structure defects such as the large holes. In addition, the research results also suggest that increasing the stretching rate higher than 80 mm/min does not change the crystallization behavior of the membranes further. 

## 4. Conclusions

In this paper, the PVDF hollow fiber microporous membrane prepared via the TIPS (L-L) process is exposed to a treatment consisting of room temperature stretching and subsequent annealing at 145 °C to induce crystal phase transformation and improve the membrane morphology and performance. First, the force field of the stretching and the thermal effect of the annealing facilitate an obvious elevation in the crystallinity of the membrane materials, as well as the generation of the β crystalline phase PVDF. At the same time, the stretching and annealing lead to the orientation of the membrane pores along the stretching direction and result in an increase in the mean pore size. The membrane performance, including the tensile strength and the PWF data, have also been improved, although the elongation at break is depressed. Compared to the stretching rate, changing the stretching ratio is more significant for the regulation of PVDF membrane structure and performance. In general, as the stretching ratio is 50% and the stretching rate is 20 mm/min, the bicontinuous pore structure of the membrane remains uniform and the mean pore size is 268.7 nm which is slightly higher than that of the original membrane sample, whereas the tensile strength is improved by 36% to 7.47 MPa, and the PWF is almost double which is much higher than the results of the literature. This research not only provides an effective way to improve and optimize the performance of bicontinuous microporous membranes prepared via the TIPS (L-L) process but also helps to understand the structure and properties of PVDF membrane materials. 

## Figures and Tables

**Figure 1 membranes-10-00038-f001:**
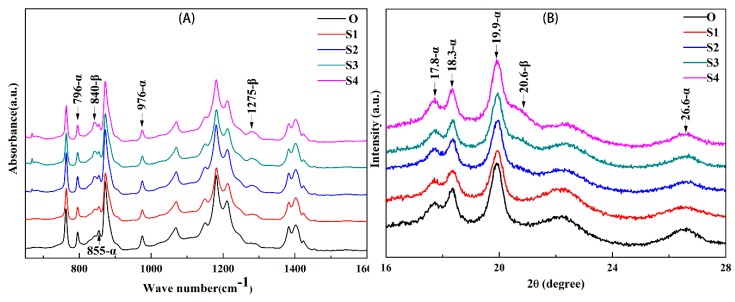
The attenuated total reflectance-Fourier transform infrared (ATR-FTIR) spectra (**A**) and X-ray diffractograms (**B**) of the membrane samples stretched with different stretching ratios.

**Figure 2 membranes-10-00038-f002:**
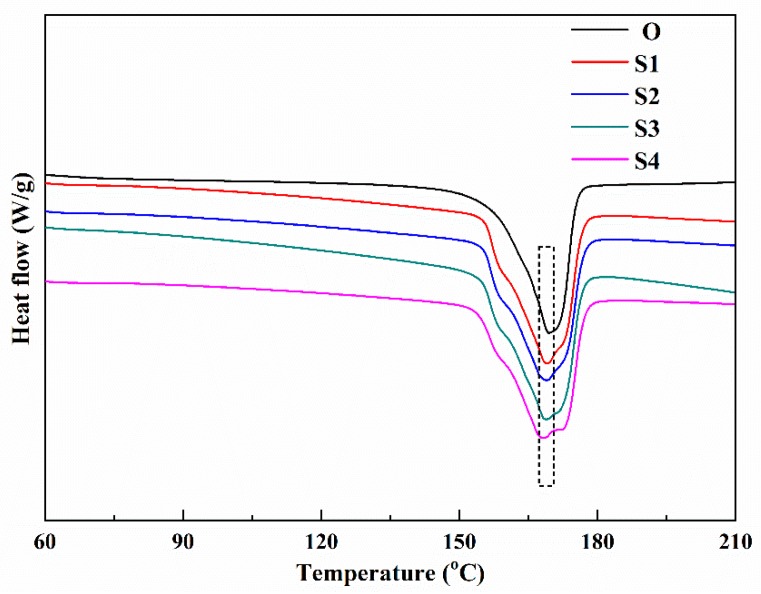
The differential scanning calorimetry (DSC) curves of the membrane samples stretched with different stretching ratios.

**Figure 3 membranes-10-00038-f003:**
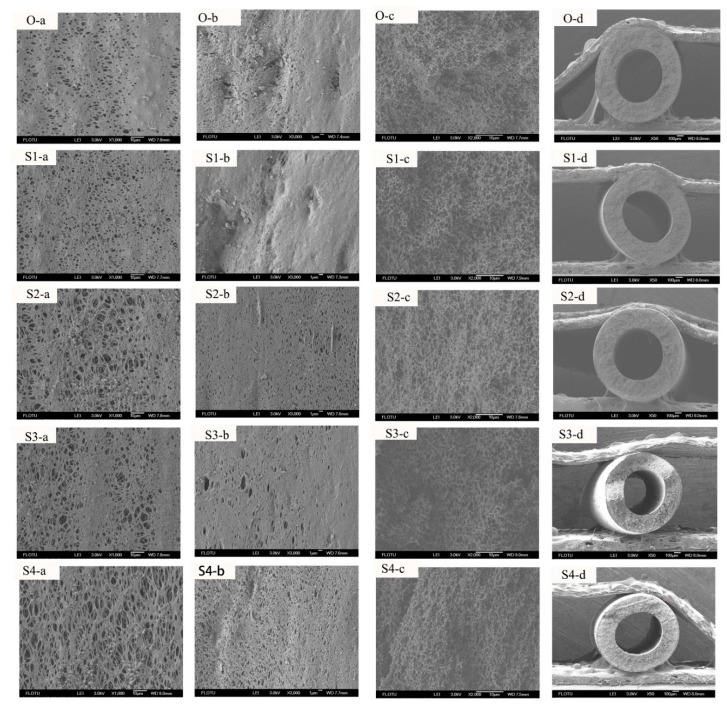
The scanning electron microscope (SEM) photos of the inner surface, the outer surface, cross-section (along the stretching direction), and the whole cross-section of the membrane samples stretched with different stretching ratios. In the figure, **a**, **b**, **c**, and **d** represent the inner surface, the outer surface, the cross-section (along the stretching direction), and the whole cross-section of the hollow fiber membrane samples, respectively.

**Figure 4 membranes-10-00038-f004:**
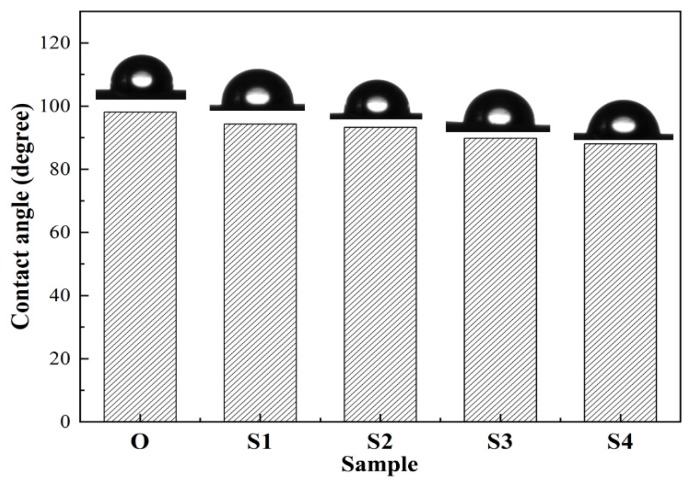
The contact angles of the membrane samples stretched with different stretching ratios.

**Figure 5 membranes-10-00038-f005:**
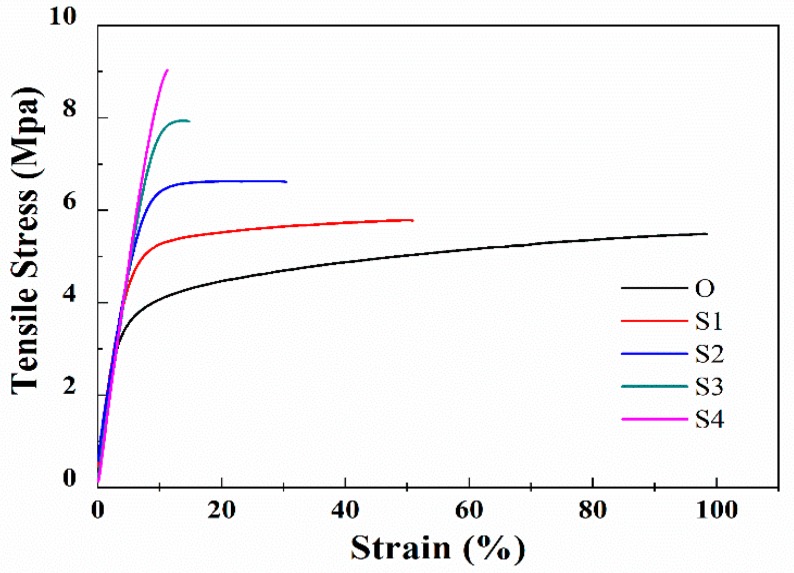
The stress-strain curves of the membrane samples stretched with different stretching ratios.

**Figure 6 membranes-10-00038-f006:**
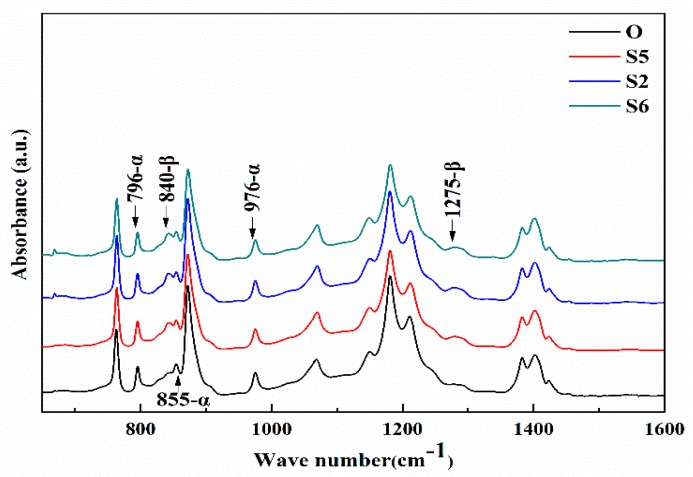
The ATR-FTIR spectra of the membrane samples stretched with different stretching rates.

**Figure 7 membranes-10-00038-f007:**
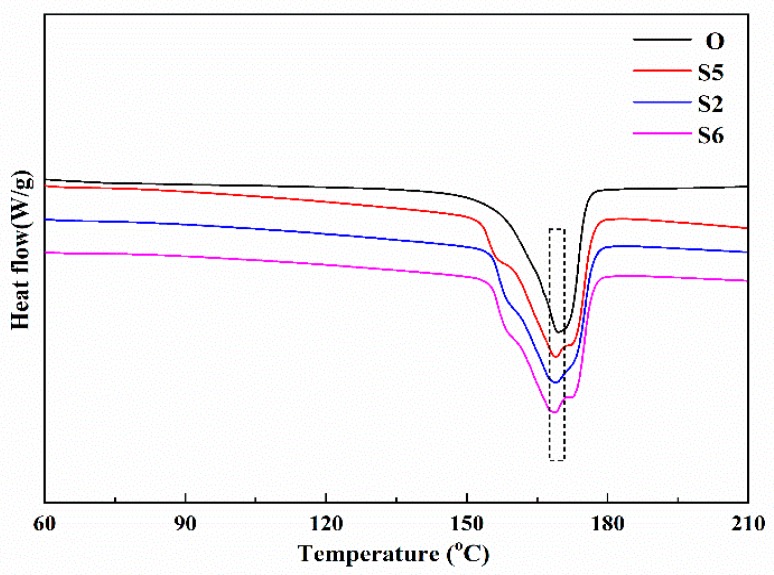
The DSC curves of the membrane samples stretched with different stretching rates.

**Figure 8 membranes-10-00038-f008:**
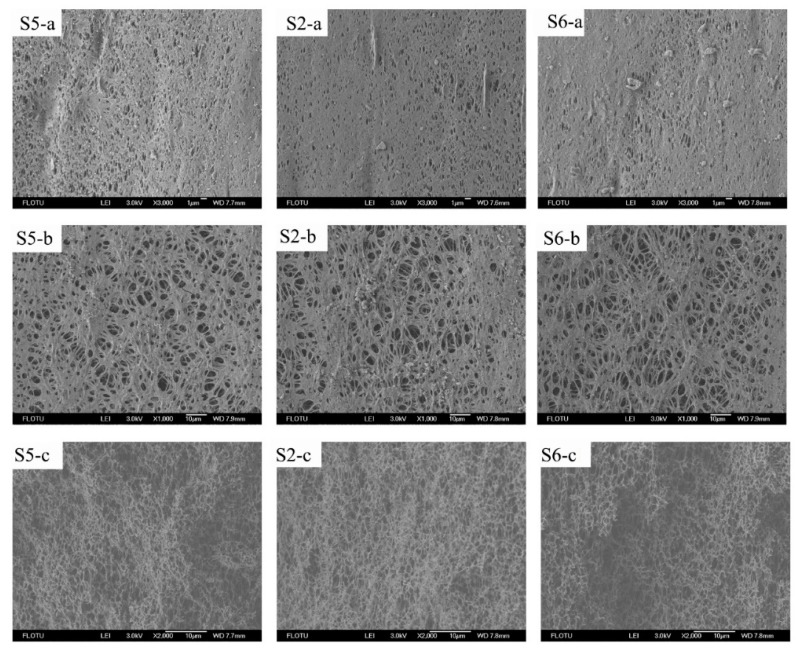
The SEM photos of the outer, inner surface, and the cross-section (along the stretching direction) of the membrane samples stretched with different stretching rates. In the figure, **a**, **b** and **c** represent the outer surface, the inner surface, and the cross-section (along the stretching direction) of the hollow fiber membrane samples, respectively.

**Figure 9 membranes-10-00038-f009:**
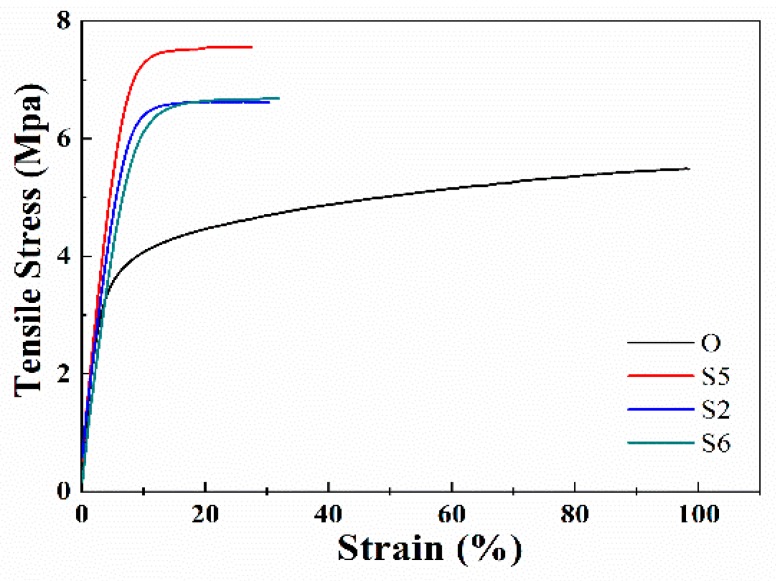
The stress-strain curves of the membrane samples stretched with different stretching rates.

**Table 1 membranes-10-00038-t001:** The melting points and total crystallinities of polyvinylidene fluoride (PVDF) hollow fiber membrane samples with different preparation conditions.

Sample	Stretching Ratio + Stretching Rate	Melting Peak Point/°C	Total Crystallinity /%
O	Original membrane	169.3	43.6
S1	20% + 40 mm/min	168.9	50.8
S2	50% + 40 mm/min	168.8	51.7
S3	80% + 40 mm/min	168.8	52.1
S4	100% + 40 mm/min	168.0	53.6
S5	50% + 20 mm/min	168.8	51.2
S6	50% + 80 mm/min	168.7	52.5

**Table 2 membranes-10-00038-t002:** The mean pore size, tensile strength, and pure water flux (PWF) of the PVDF hollow fiber membrane samples.

Sample	Mean Pore Size/nm	Tensile Strength/MPa	Pure Water Flux/L·m^−2^·h^−1^·(0.1bar)^−1^
O	235.4	5.48	457.01
S1	249.6	5.73	594.51
S2	289.2	6.56	776.28
S3	326.7	7.83	997.60
S4	373.4	9.01	1000.22
S5	268.7	7.47	796.13
S6	283.4	6.32	771.54

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
