# Peer review of "Effects of Room Temperature Stretching and Annealing on the Crystallization Behavior and Performance of Polyvinylidene Fluoride Hollow Fiber Membranes"

_membranes, 2020, doi:10.3390/membranes10030038_

Round 1

Reviewer 1 Report

  1. Minor revision of English: line 63 fewer than?
  2. line 53, what is cooperative stretching process, maybe would worth to describe it.

Author Response

Response to Reviewer 1 Comments

We are very grateful to the reviewer for the valuable comments and suggestions. Our manuscript has been revised accordingly and the responses to all comments are summarized as follows. Overall, all changes based on the comments have been marked in red in the revision.

Reviewer 1

Point 1: Minor revision of English: line 63 fewer than?

Response 1: In the revised paper, the “fewer” has been changed into “few” according to the suggestion.

Point 2: In Line 53, what is cooperative stretching process, maybe would worth describing it.

Response 2: In the literature, the “cooperative stretching process” means that the PVDF thin films were coated on stretchable poly (vinyl alcohol) substrates, and then they were cooperatively stretched under optimized stretching parameters. Therefore based on this advice, the sentence in Line 52 has been replaced by “Zhu et al. found that by a cooperative stretching process of PVDF films deposited on stretchable poly (vinyl alcohol) substrates, the PVDF molecules could be oriented along the direction of stretching and the nonpolar α phase was converted to electroactive β and γ phases with an increase of stretching ratio from 1 to 3”.

Reviewer 2 Report

General comments:

1. The “stretching ratio” should be explained in the introduction

2. Please explain what “improving the structure” (line 87) mean? It is too general statement. What is the main aim of PVDF membrane modification, what results are expected?

3. Explain why 145°C and 15 min were chosen as annealing temperature and time

4. As the membrane surface changes with the stretching ratio, is it possible to also affect the contact angle measurements (see Figure 4 and the discussion provided)? It is well known that the surface roughness and porosity can strongly influence the contact angle values.

5. In Introduction Authors write: “The results have shown that the mechanical performance of the PVDF membranes could be improved by increasing the membrane crystallinity as well as the β crystalline phase content [6-8].” and “They found that stretching ratio 84 positively affected the permeability and porosity but decreased the fiber strain and stretching the 85 fibers up to 40% wasn’t enough to induce any detectable the β phase crystals.” – please referee to these statements in mechanical properties discussion. Why results presented in [6-8] differ from those obtained by Authors and others?

Particular comments:

6. Page 1 line 35: change “property” into “properties”

7. Page 2, line 46: nanoparticles where rather “additive” than “addition”

8. Page 2, line 47: I suggest not to write “improved” but “changed” as for different applications improvement can be regarded in a different way

9. Page 3, line 95: “Materials and PVDF hollow fiber microporous membranes:” – remove

10. Page 3, line 113: “change “crystallization phase” into “crystalline phase”

11. Page 3, line 116: the samples were heated to 220 (even if melting takes place during heating)

12. Explain how “melting data” was used – give exact value used for calculation

13. Page 4, line 158: “The spectra…” – in Fig.1B not spectra but diffractograms are shown

14. Page 4, line 164: in my opinion, this peak is also hardly observed for S1 sample

15. Page 4, line 175: “while a small melting peak near 157 °C” – add info that for DSC-curves of S1-S4

16. Line 178: correct: “higher melting peak”

17. Line 207: “in accord….”?

18. Line 304: “change “section 2.1” to “section 3.1”

Others:

19. Page 2, line 67 and 69, style [14], [9,15-17] – remove superscript

Author Response

Response to Reviewer 2 Comments

We really appreciate the Reviewer for the valuable comments and suggestions. Our manuscript has been revised accordingly and the responses are summarized as follows. In general, all changes based on the comments have been marked in red in the revised manuscript.

Point 1: The “stretching ratio” should be explained in the introduction.

Response 1: Thank the reviewer for the advice, and then a new sentence was added in Line 94 to explain the ‘stretching ratio’ as ‘It should be noted that herein the stretching ratio indicates the length ratio of the stretched membranes to the original ones’.

Point 2: Please explain what “improving the structure” (line 87) mean? It is too general statement. What is the main aim of PVDF membrane modification, what results are expected?

Response 2: According to the suggestion, the phase ‘improving the structure’ in Line 88 has been revised into ‘improving the permeability and mechanical strength’. We expect to prompt the permeability and mechanical strength of the membranes by the modification.

Point 3: Explain why 145°C and 15 min were chosen as annealing temperature and time.

Response 3: In this work, the annealing was applied to release the stress, otherwise the membrane samples would shrink back as soon as the clamps were opened. We chose the annealing temperature to be 145 oC, since it was near the onset crystallization temperature of PVDF, around 150 oC according to Fig.2. The annealing time was chosen to be 15 minutes due to some experimental tests. We found that basically after 15 minutes, the membrane samples wouldn’t shrink back anymore. Herein the revised version, we have added a sentence ‘that is near the onset crystallization temperature of PVDF, around 150 oC,’ in Line 104.

Point 4: As the membrane surface changes with the stretching ratio, is it possible to also affect the contact angle measurements (see Figure 4 and the discussion provided)? It is well known that the surface roughness and porosity can strongly influence the contact angle values.

Response 4: We do agree with the reviewer on this view that the surface roughness and porosity can strongly influence the contact angle values. As shown in Fig.3, the outer surface structure of Sample S1 is similar to that of Sample O due to the stretching ratio is only 20%, nevertheless, the contact angle of Sample S1 is also smaller than that of Sample O. That’s why we think the reason for the decrease of the contact angle should be attributed to the formation of the β crystalline phase of PVDF. In addition, according to the suggestion, we have added more explanation to Line 231.

Point 5: In introduction, authors write: “The results have shown that the mechanical performance of the PVDF membranes could be improved by increasing the membrane crystallinity as well as the β crystalline phase content [6-8].” and “They found that stretching ratio positively affected the permeability and porosity but decreased the fiber strain and stretching the fibers up to 40% wasn’t enough to induce any detectable the β phase crystals.” – please referee to these statements in mechanical properties discussion. Why results presented in [6-8] differ from those obtained by Authors and others?

Response 5: We appreciate the suggestion very much. We think that the results from references [6-8] agree with the observation obtained by us and the other researchers. Maybe the statement in the manuscript caused this misunderstanding. According to our research, increasing the stretching ratio really has a positive impact on the tensile strength, the porosity, and the permeability but a negative effect on the strain (elongation at break). Considering the ATR-FTIR spectra and the X-ray diffractograms of Fig.1, the formation of β crystalline phase of PVDF is not obvious when the stretching ratio is lower than 50% and becomes evident as the stretching ratio is higher, which also agrees with the conclusion obtained from the literature [21]. According to the advice, we have added more discussion in Line 254 to make our points clear.

English language edits:

Particular comments:

Point 6: Page 1 line 35: change “property” into “properties”

Response 6: The word “property” has been changed into “properties” in the revised version.

Point 7: Page 2, line 46: nanoparticles where rather “additive” than “addition”

Response 7: The word “addition” has been modified into“additives” in the revised version.

Point 8: Page 2, line 47: I suggest not to write “improved” but “changed” as for different applications improvement can be regarded in a different way.

Response 8: According to the reviewer, the words “the nanoparticle improved” in Line 47 were revised into “the nanoparticle montmorillonite improved”.

Point 9: Page 3, line 95: “Materials and PVDF hollow fiber microporous membranes:” – remove

Response 9: In the revised version, the words“Materials and PVDF hollow fiber microporous membranes:” in Line 97 have been deleted.

Point 10: Page 3, line 113: “change “crystallization phase” into “crystalline phase”

Response 10: The words “crystallization phase” in Line 115 have been changed into “the crystalline phase”.

Point 11: Page 3, Line 118: the samples were heated to 220 (even if melting takes place during heating)

Response 11: The sentence in Line 118 has been revised into “The membrane samples were melted to 220°C (even if melting takes place during heating) at 10°C/min and then cooled to 40°C”.

Point 12: Explain how “melting data” was used – give exact value used for calculation

Response 12: This time, we provide more information from Line 119 to Line 222 to explain how the “melting data” was used.

Point 13: Page 4, line 158: “The spectra…” – in Fig.1B not spectra but diffractograms are shown.

Response 13: Thank the reviewer for the consideration. The word “spectra” in Line 169 has been replaced by “X-ray diffractograms” in the revised version.

Point 14: Page 4, line 164: in my opinion, this peak is also hardly observed for S1 sample.

Response 14: Thank the reviewer for the worthy suggestion. This time, the corresponding paragraph has been rewritten to make the description clear and accurate. The corresponding revised parts have been marked in red.

Point 15: Page 4, line 175: “while a small melting peak near 157 °C” – add info that for DSC-curves of S1-S4.

Response 15: Thanks for the valuable suggestion, here in the revised version, the words “for Sample S1 to S4,” have been added in Line 188.

Point 16: Line 178: correct: “higher melting peak”

Response 16: According to the advice, the words “higher melting peak” have been changed into “another melting peak appears on the right side of the dashed box” in Line 193.

Point 17: Line 207: “in accord….”?

Response 17: According to the suggestion, the words “are in accord with …” in Line 221 have been revised into “agrees with”.

Point 18: Line 304: “change “section 2.1” to “section 3.1”

Response 18: The words mentioned by the reviewer now in Line 328 have been changed into “section 3.1”.

Point 19: Page 2, line 67 and 69, style [14], [9,15-17] – remove superscript

Response 19: The superscript styles mentioned by the Reviewer have all been removed.

Round 2

Reviewer 2 Report

I have only one suggestion:

 “The membrane samples were melted to 220°C (even if melting takes 119 place during heating) at 10°C/min and then cooled to 40°C.” change into “The membrane samples were heated to 220°C at 10°C/min and then cooled to 40°C.”

Author Response

Response to Reviewer 2 Comments

We are very grateful to the reviewer for the valuable comment. Our manuscript has been revised accordingly and the response is listed as follows. The change based on the comment has been marked in blue to be different from the first revisions in the manuscript.

Reviewer 2

Point 1: “The membrane samples were melted to 220°C (even if melting takes place during heating) at 10°C/min and then cooled to 40°C.” change into “The membrane samples were heated to 220°C at 10°C/min and then cooled to 40°C.”

Response 1: In the revised paper, the sentence has been replaced by “The membrane samples were heated to 220°C at 10°C/min and then cooled to 40°C.”